# *Hedera rhombea* inhibits the biofilm formation of *Candida*, thereby increases the susceptibility to antifungal agent, and reduces infection

Daseul Kim[1], Ki-young Kim[1,2]*

1 Graduate School of Biotechnology, Kyung Hee University, Seocheon, Giheung, Yongin, Gyeonggi-do, Republic of Korea, 2 College of Life Science, Kyung Hee University, Seocheon, Giheung, Yongin, Gyeonggi-do, Republic of Korea

* kiyoung@khu.ac.kr

**Data Availability Statement:** All relevant data are within the paper and its Supporting Information files.

## Abstract

*Candida* is an opportunistic pathogen and a common cause of fungal infections worldwide. Anti-fungal use against *Candida* infections has resulted in the appearance of resistant strains. The limited choice of anti-fungal therapy means alternative strategies are needed to control fungal infectious diseases. The aim of this study was to evaluate the inhibition of *Candida* biofilm formation by *Hedera rhombea* (Korean name: songak) extract. Biofilm formation was assessed using the crystal violet assay which showed a dose dependent reduction in the presence of extract with the biofilm formation inhibitory concentration of *C. albicans* ($IC_{50}$ = 12.5μg/ml*)*, *C. tropicalis var. tropicalis* ($IC_{50}$ = 25μg/ml), *C. parapsilosis var. parapsilosis* ($IC_{50}$ = 6.25μg/ml), *C. glabrata* ($IC_{50}$ = 6.25μg/ml), *C. tropicalis* ($IC_{50}$ = 12.5μg/ml), and *C. parapsilosis* ($IC_{50}$ = 12.5μg/ml) without directly reducing *Candida* growth. Treatment with 6.25μg/mL of extract increased the antifungal susceptibility to miconazole from 32% decreasing of fungal growth to 98.8% of that based on the fungal growth assay. Treatment of extract dose-dependently reduced the dimorphic transition of *Candida* based on the dimorphic transition assay and treatment of 3.125μg/mL of extract completely blocked the adherence of *Candida* to the HaCaT cells. To know the molecular mechanisms of biofilm formation inhibition by extract, qRT-PCR analysis was done, and the extract was found to dose dependently reduce the expression of hyphal-associated genes (*ALS3*, *ECE1*, *HWP1*, *PGA50*, and *PBR1*), extracellular matrix genes (*GSC1*, *ZAP1*, *ADH5*, and *CSH1*), Ras1-cAMP-PKA pathway genes (*CYR1*, *EFG1*, and *RAS1*), Cph2-Tec1 pathway gene (*TEC1*) and MAP kinases pathway gene (*HST7*). In this study, *Hedera rhombea* extract showed inhibition of fungal biofilm formation, activation of antifungal susceptibility, and reduction of infection. These results suggest that fungal biofilm formation is good screen for developing the antifungal adjuvant and *Hedera rhombea* extract should be a good candidate against biofilm-related fungal infection.

**Funding:** This research was funded by the GRRC Program of Gyeonggi province [GRRC-KyungHee2020(B04)], Republic of Korea.

**Competing interests:** The authors have declared that no competing interests exist.

## Introduction

*Candida albicans* is an opportunistic pathogen which is responsible for systemic infections in immunocompromised patients. *C. albicans* can persist inside the host and can be aided by drug resistance traits which often lead to failure of therapeutic strategies [1]. One of the features of *Candida* species pathogenesis is their ability to form biofilms, and nosocomial infections are often related to the ability to produce biofilm on mucosal surfaces and implanted medical devices [2–5].

The formation of biofilms involves multiple interconnected signaling pathways [6–14], and is a finely controlled process that involves attachment to surface and embedment in the exopolymer extracellular matrix [15–18]. The biofilm matrix acts to structure microbial communities and includes sessile cells that are frequently much more resistant to antifungal agents. In fact, adherent *C. albicans* cells without specific drug resistant gene expression are up to 1,000 times more resistant to common antifungal agents than planktonic cells [19]. Therefore, the biofilm of *C. albicans* is a reservoir of viable fungal cells that can potentially cause systemic infections, with a mortality rate of around 40–60% [20, 21]. Efforts are being made to develop alternative strategies to eradicate biofilm-related infections [22, 23]. Medicinal plants are used for diverse traditional methods to treat cancer, infection, fever, asthma, and many other diseases. Herbal medicines usually have fewer side effects compared to over-the-counter medicines. Accordingly, medicinal plants should be a new provenance of replacement remedies to treat *Candida* infectious diseases [24–26].

*Hedera rhombea* is a species of ivy (genus Hedera) that is native to the coast and some islands of East Asia [27, 28]. In oriental medicine, *H. rhombea* is mainly used for arthritis, low back pain, hepatitis, high blood pressure, hemostasis, anti-rheumatism, facial paralysis, jaundice anti-inflammatory action, hypertension, and antitumor [27–29].

In this study, *H. rhombea* extract showed anti-biofilm formation activity against several fungi including *C. albicans*, *C. tropicalis*, *C. glabrata*, and *C. parapsilosis*. Interestingly, the activity of the extract also increased susceptibility to antibiotics. These results suggest that *H. rhombea* extract can be used as a potential anti-fungal adjuvant to control the biofilm-related infection.

## Materials and methods

### Strains

The *Candida* strains used in this study are listed in Table 1. All strains were stored in 20% glycerol at −70˚C and cultured in YPD plates [peptone 20 g/L (BD Difco, Belgium), yeast extract 10 g/L (BD Difco, Belgium) and 2% glucose (w/v) (Daejung, Korea)].

**Table 1. Strains used in this study.**

| Strain | Description | Source |
|---|---|---|
| *C. albicans* | KCTC 7965 | Purchased form KCTC (Korean Collection for Type of Cultures) or KACC (Korean Agriculture Culture Collection) |
| *C. tropicalis var. tropicalis* | KCTC17762 | |
| *C. parapsilosis var. parapsilosis* | KACC45480 | |
| *C. glabrata* | KCTC7219 | |
| *C. tropicalis* | KCTC7212 | |
| *C. parapsilosis* | KACC49573 | |

## *H. rhombea* extraction

The leaf of *H. rhombea* was obtained from Jeju island. The extracts were produced using distilled water in 3L containing 300g of the sample at 80°C for 8 h, concentrate at 40°C using a rotary evaporator, and freeze-dried. 10mg of plant extract powder was dissolved in 1mL dimethyl sulfoxide for the experiments [15, 24–28, 30].

## Inhibition of biofilm formation

*H. rhombea* extract ranging from 6.25 to 100μg/mL were prepared in 96-well flat-bottomed plates (SPL, Korea). Wells without test compounds served as controls (DMSO concentration of 0.1%). $1 \times 10^6$ CFU/mL of *C. albicans* suspension were prepared in RPMI 1640 medium [26, 31–34, 36–40]. Then 100μl of the solution was inoculated into 96-well flat-bottomed plates. After incubation at 37°C for 24h, non-adherent cells were removed by washing with PBS and then 100ul of 1% aqueous crystal violet was applied for 30 minutes. Each well was washed three times with PBS and instantly de-stained with 150μl of 30% acetic acid for 15min. The absorbance was measured at 595nm with a microplate reader (Bio Tek Instruments, Korea). The experiments were performed in triplicate (Fig 1).

## Combinatorial antifungal effects of *H. rhombea* extract with antifungal agents

*C. albicans* were grown overnight in YPD diluted to $1 \times 10^6$ cells/mL. The induction of biofilm formation was performed as described above. Miconazole (3.125μg/mL), magnoflorine (3.125μg/mL) and dioscin (3.125μg/mL) alone or with the indicated concentration of *H. rhombea* extract were added and incubated at 37°C for 24 h [26, 30–34]. The experiments were performed in triplicate.

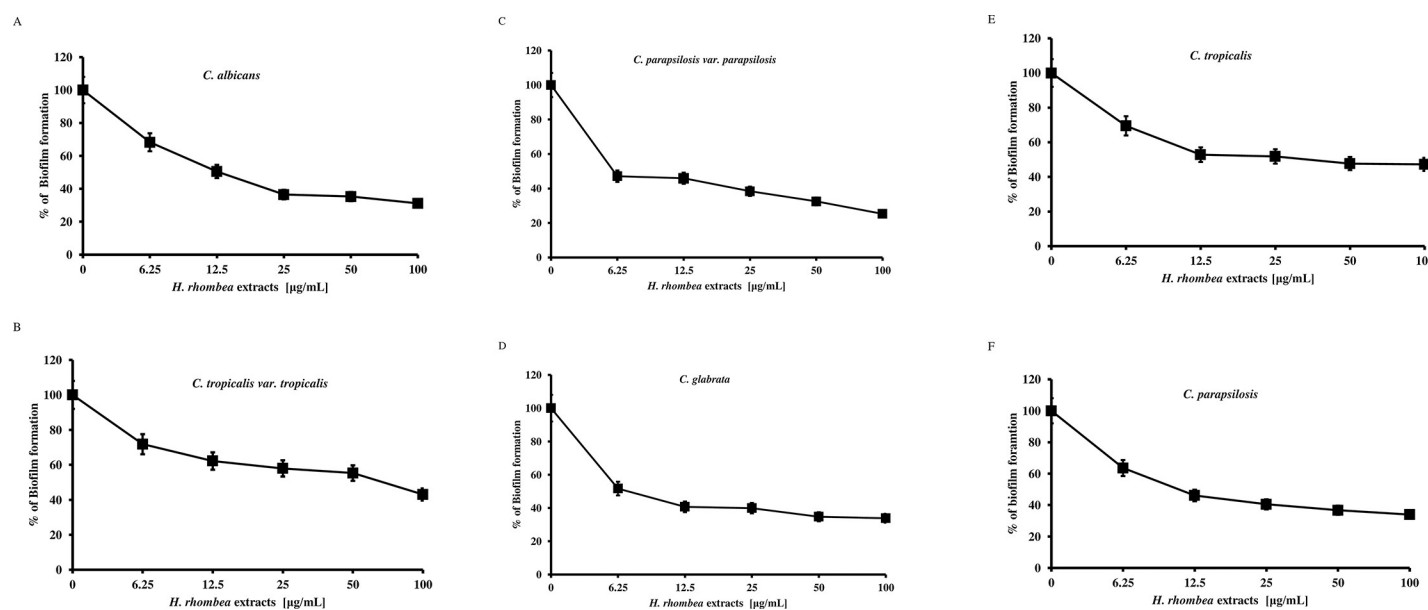

**Fig 1. Inhibition of *Canddia* spp. biofilm formation by *H. rhombea* extract.** The biofilm formation of (A) *C. albicans*, (B) *C. tropicalis var. tropicalis*, (C) *C. parapsilosis var. parapsilosis*, (D) *C. glabrata*, (E) *C. tropicalis*, and (F) *C. parapsilosis* was induced in YPD media supplemented with 10% fetal bovine serum with the indicated concentrations of *H. rhombea* extract at 37°C for 24h.

## Dimorphic transition of *C. albicans*

*C. albicans* were grown overnight in YPD medium. $1 \times 10^6$ cells/mL of *Candida* with or without extracts were incubated in RPMI 1640 medium and YPD media supplemented with 10% fetal bovine serum medium to induce dimorphic transition at 37˚C for 4 h. RPMI without sodium barcarbonate and with glutamine buffered with MOPS [3-(N-morpholino) propane-sulfonic acid] to pH 7. Inhibition quantification of the yeast-to-hyphal-form transition was accomplished by counting the number of hyphae cells in the population as previously described [26, 32–40]. More than 1,000 cells were counted for each well in duplicate, and all assays were repeated five times. Representative results of images were obtained using a fluorescence microscope (EVOS® FL, ThermoFisher Scientific, Waltham, MA, USA). The experiments were performed in triplicate.

## *Candida* adherence test

A human epithelial keratinocyte HaCaT cells were maintained in DMEM supplemented with 10% fetal bovine serum at 37˚C in a humidified atmosphere of 5% $CO_2$.

HaCaT cells ($0.5 \times 10^6$ cells per wall) were grown to confluence on 24 well plates for 24 h. DMEM were drained and then plates were cautiously washed three times with PBS to remove nonadherent cell. $1 \times 10^6$ cell/mL *C. albicans* mixed with *H. rhombea* extract ranged 1.56–25μg/mL concentration was treated into each well. The 24 well plates were incubated at 37˚C for 24 h. Representative results of images were obtained using a microscope [32, 34]. The experiments were performed in triplicate.

## qRT-PCR analysis

*C. albicans* was grown overnight in YPD and diluted to $1 \times 10^6$ cells/mL. The diluted suspension with 6.25–100μg/mL of *H. rhombea* extract was incubated RPMI 1640 at 37˚C for 24 h with shaking. Total RNA was isolated using TRIzol reagent (Life Technology, Thermo Fisher Scientific, USA) according to the manufacturer's instruction and the reverse transcriptase (NanoHelix, Korea) reaction was prepared using 1 μg of RNA to obtain cDNA. qRT-PCR was carried out with the 2X SybrGreen qPCR Mater Mix (CellSafe, Korea). The transcript level of detected genes was calculated using the formula $2^{-\Delta\Delta CT}$. Primer sequences used are listed in Table 2. *ACT1* was used as internal control [26, 30–34, 38–40]. The experiments were performed in triplicate.

## MTT assay

The cytotoxicity of *H. rhombea* extract against HaCaT and THP-1 were tested by a slightly modified MTT assay [26, 32–34]. Briefly, $1 \times 10^4$ HaCaT and THP-1 in DMEM and RPMI1640 medium, respectively were added to each well containing the indicated concentration of extract and incubated for 24 h. Cell viability was calculated by optical density ($OD_{540}$) values measured using a microplate reader (BioTek Instruments, Korea) and is reported as the percentage of the vehicle control [32]. The experiments were performed in triplicate.

## Growth inhibition assay for *C. albicans*

Fungal culture was prepared with the fresh YPD to $1 \times 10^6$ cells/mL of *C. albicans* [30–33, 40]. 100μg/mL of *H. rhombea* extract was added and then incubated at 37˚C. The growth was evaluated by measuring $OD_{600}$ using microplate reader after 0, 1, 2, 4, 8, 12, and 24 h [30–32]. The experiments were performed in triplicate.

**Table 2. Primers for *C. albicans* genes used in this study.**

| Genes | Primer sequence | Gene function | References |
|---|---|---|---|
| ACT1 | F: TAGGTTTGGAAGCTGCTGG | Control | [10] |
| | R: CCTGGGAACATGGTAGTAC | | |
| CAN2 | F: GCGGAATGGATATGCATGGG | Biofilm formation | In this study |
| | R: CGGATTGCTCTTGGAGAAGC | | |
| EHT1 | F: TCGGAAAGCTTGGTGAAAGC | Biofilm formation | In this study |
| | R: ATTTGGCCAAAGCAGGACTC | | |
| TPO4 | F: GCGGAATGGATATGCATGGG | Biofilm formation | In this study |
| | R: CGGATTGCTCTTGGAGAAGC | | |
| OPT7 | F: TTGATCCCAGCTGCCAAATG | Biofilm formation | In this study |
| | R: TGGCCCAAGTTCTTCGTATC | | |
| CYR1 | F: GTTTCCCCCACCACTCA | Ras1-cAMP-Efg1 pathway | [10] |
| | R: TTGCGGTAATGACACAACAG | | |
| EFG1 | F: TTGAGATGTTGCGGCAGGAT | Ras1-cAMP-Efg1 pathway | [10] |
| | R: ACTGGACAGACAGCAGGAC | | |
| HST7 | F: GCCAGTATGGTCGGAGGAT | MAP kinases pathway | [10] |
| | R: ACATAGGCATCGTCTTCGTC | | |
| RAS1 | F: GAGGTGGTGGTGTTGGTA | Ras1-cAMP-Efg1 pathway | [10] |
| | R: TCTTCTTGTCCAGCAGTATC | | |
| TEC1 | F: GCACTGGCTTCAAGCTCAAA | Extracellular matrix | [10] |
| | R: GCTGCTGCACCAAGTTCTG | | |
| ALS3 | F: GGTTATCGTCCATTTGTTG | Hyphal-specific genes | [10] |
| | R: TTCTGTATCCAGTCCATCT | | |
| ECE1 | F: ACAGTTTCCAGGACGCCAT | Hyphal-specific genes | [10] |
| | R: ATTGTTGCTCGTGTTGCCA | | |
| HWP1 | F: ACAGGTAGACGGTCAAGG | Ras1-cAMP-Efg1 pathway | [10] |
| | R: GGGTAATCATCACATGGTTC | | |
| PBR1 | F: TGTTGCTGCTGGTTCTGATG | Hyphal-specific genes | In this study |
| | R: GGTGGCAGATTTGGATTACC | | |
| PGA50 | F: ATTTCGAAGGGTGCAACTGC | Hyphal-specific genes | In this study |
| | R: AAGCACTGCAATGGGAGTTG | | |
| ADH5 | F: ACCTGCAAGGGCTCATTCTG | Extracellular matrix | [10] |
| | R: CGGCTCTCAACTTCTCCATA | | |
| CSH1 | F: CGTGAGGACGAGAGAGAAT | Extracellular matrix | [10] |
| | R: CGAATGGACGACACAAAACA | | |
| GSC1 | F: CCCATTCTCTAGGCACGA | Extracellular matrix | [10] |
| | R: ATCAACAACCACTTGCTTCG | | |
| ZAP1 | F: ATCTGTCCAGTGTTGTTTGTA | Extracellular matrix | [10] |
| | R: AGGTCTCTTTGAAAGTTGTG | | |

## Statistical analysis

All experiments were performed at least three times and data were presented as the ± mean S.D.

## Result

### Inhibition of *Candida* biofilm formation by the treatment of *H. rhombea* extract

Biofilm is especially important for the fungi to survive and infect. *H. rhombea* was used to test whether it blocked fungal biofilm formation or not. *H. rhombea* extract dose-dependently

inhibited the *Candida* biofilm formation in all the tested strains (Table 1) with the $IC_{50}$ value of approximately 6.25μg/mL for *C. albicans* (Fig 1A).

$IC_{50}$ values of other strains were 6.25μg/mL (*C. parapsilosis* var. *parapsilosis* and *C. glabrata*), 12.5μg/mL (*C. tropicalis* and *C. parapsilosis*), and 25μg/mL (*C. tropicalis* var. *tropicalis*) (Fig 1B, 1C, 1D, 1E and 1F).

## *H. rhombea* extract increased the susceptibility to an antifungal agent against *C. albicans*

*H. rhombea* extract increased the susceptibility of antifungal agents against *C. albicans*. Treatment of 3.125μg/mL of extract increased the susceptibility to miconazole with 99% of fungal growth inhibition from 38% inhibition by miconazole treatment. The extract also increased the susceptibility to plant-derived antifungal candidates including magnoflorine (99% growth inhibition by 6.25μg/mL of extract compared with 29% growth inhibition by 3.125ug/ml of only magnoflorine treatment) and dioscin (99% growth inhibition by 6.25μg/mL of extract compared with 39% growth inhibition by 3.125ug/ml of only dioscin treatment) (Fig 2A, 2B and 2C).

## *H. rhombea* extract blocked dimorphic transition from yeast to hyphae form

The yeast-to-hyphae conversion is an important virulence property of *C. albicans*. The formation of hyphae aids the subsequent invasive growth of *C. albicans* to penetrate host tissues and lead to the establishment of systemic infection [7]. Extract tested whether the dimorphic transition was influenced and found 1.56μg/mL of the extract significantly inhibited hyphae formation in RPMI 1640 or a 10% FBS YPD medium, and extract with higher than 6.25μg/mL completely blocked the hyphae formation (Fig 3A, 3B, 3C and 3D).

## *H. rhombea* extract reduced fungal adherence to the HaCaT cells

Fungal biofilm formation on device-associated infection is an important medical problem. *H. rhombea* extract showed dose-dependently reduced adhesion of the fungi to the human HaCaT cells with 90% reduction by treatment of 1.56μg/mL of extract (Fig 4).

## *H. rhombea* extract inhibited the expression of biofilm formation and infection related genes

To understand the molecular basis of *H. rhombea* extract of inhibition of biofilm formation and thereby reducing the infection of *Candida*, the expression of genes related to biofilm formation, hyphae growth, and cell adhesion was tested by qRT-PCR. The expression of biofilm formation related genes [*CAN2* ($IC_{50}$ = 1.56μg/mL), *EHT1* ($IC_{50}$ = 1.56μg/mL), *TPO4* ($IC_{50}$ = 1.56μg/mL), and *OPT7* ($IC_{50}$ = 1.56μg/mL)], Ras1-cAMP-PKA pathway related genes [*RAS1* ($IC_{50}$ = 1.56μg/mL), *EFG1* ($IC_{50}$ = 1.56μg/mL), *TEC1* ($IC_{50}$ = 3.125μg/mL), *HST7* ($IC_{50}$ = 3.125μg/mL), and *CYR1* ($IC_{50}$ = 1.56μg/mL)], hyphal-specific genes [*ALS3* ($IC_{50}$ = 1.56μg/mL), *ECE1* ($IC_{50}$ = 3.125μg/mL), and *HWP1* ($IC_{50}$ = 3.125μg/mL)] and extracellular matrix-related genes [*GSC1* ($IC_{50}$ = 1.56μg/mL), *ADH5* ($IC_{50}$ = 3.125μg/mL), and *CSH1* ($IC_{50}$ = 1.56μg/mL)] were significantly decreased by treatment of extract (Fig 5A, 5B, 5C and 5D).

Total RNA was extracted from *C. albicans* treated with the indicated concentration of *H. rhombea* extract using RNA extraction kit, converted to cDNA, and analyzed by qPCR with the respective primers.

A

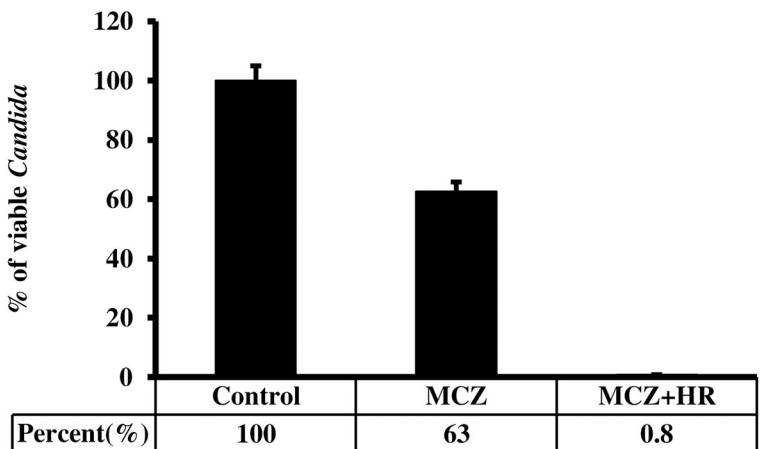

| Percent(%) | Control | MCZ | MCZ+HR |
|---|---|---|---|
| | 100 | 63 | 0.8 |

B

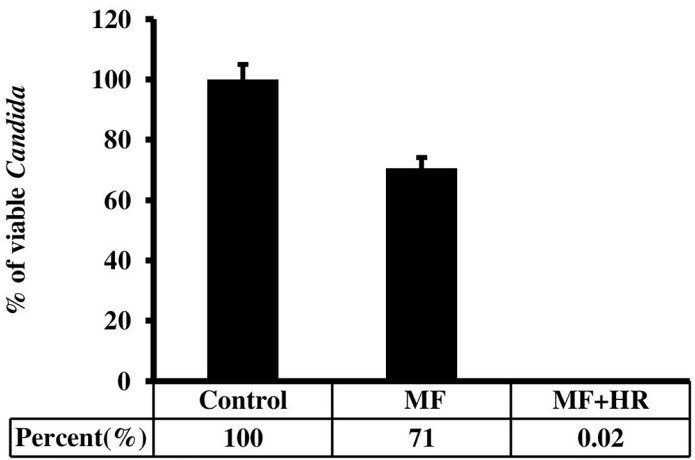

| Percent(%) | Control | MF | MF+HR |
|---|---|---|---|
| | 100 | 71 | 0.02 |

C

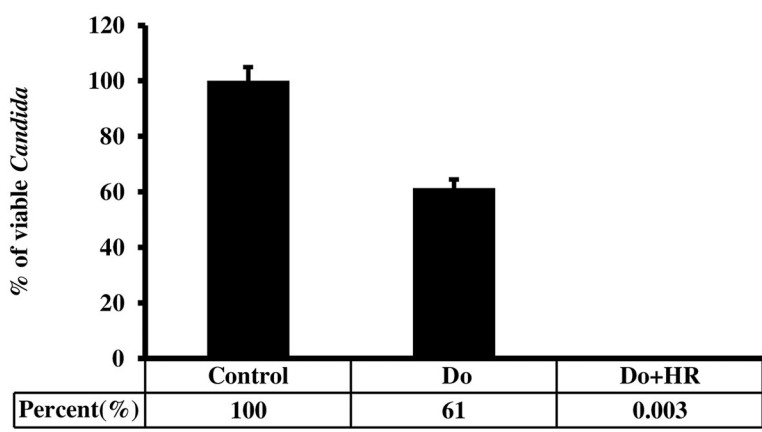

| Percent(%) | Control | Do | Do+HR |
|---|---|---|---|
| | 100 | 61 | 0.003 |

**Fig 2. *H. rhombea* extracts increased the susceptibility of *C. albicans* to miconazole etc.** *H. rhombea* extract (6.25μg/mL) increased the susceptibility to miconazole (3.125μg/mL) (A), magnoflorine (3.125μg/mL) (B) and dioscin (3.125μg/mL) (c) against *C. albicans*. The biofilm formation was induced for 24 h by growing of *C. albicans* in YPD media supplemented with 10% fetal bovine serum. MCZ: Miconazole, MF: Magnoflorine, Do: Dioscin, HR: H. rhombea extract.

## *H. rhombea* extract did not affect the growth of human originated cell

The cytotoxic effects of *H. rhombea* extract on HaCaT cells and macrophage THP-1 were checked using an MTT assay. *H. rhombea* extract has no significant cytotoxic effect on both macrophage THP-1 and HaCaT cells (Fig 6A and 6B).

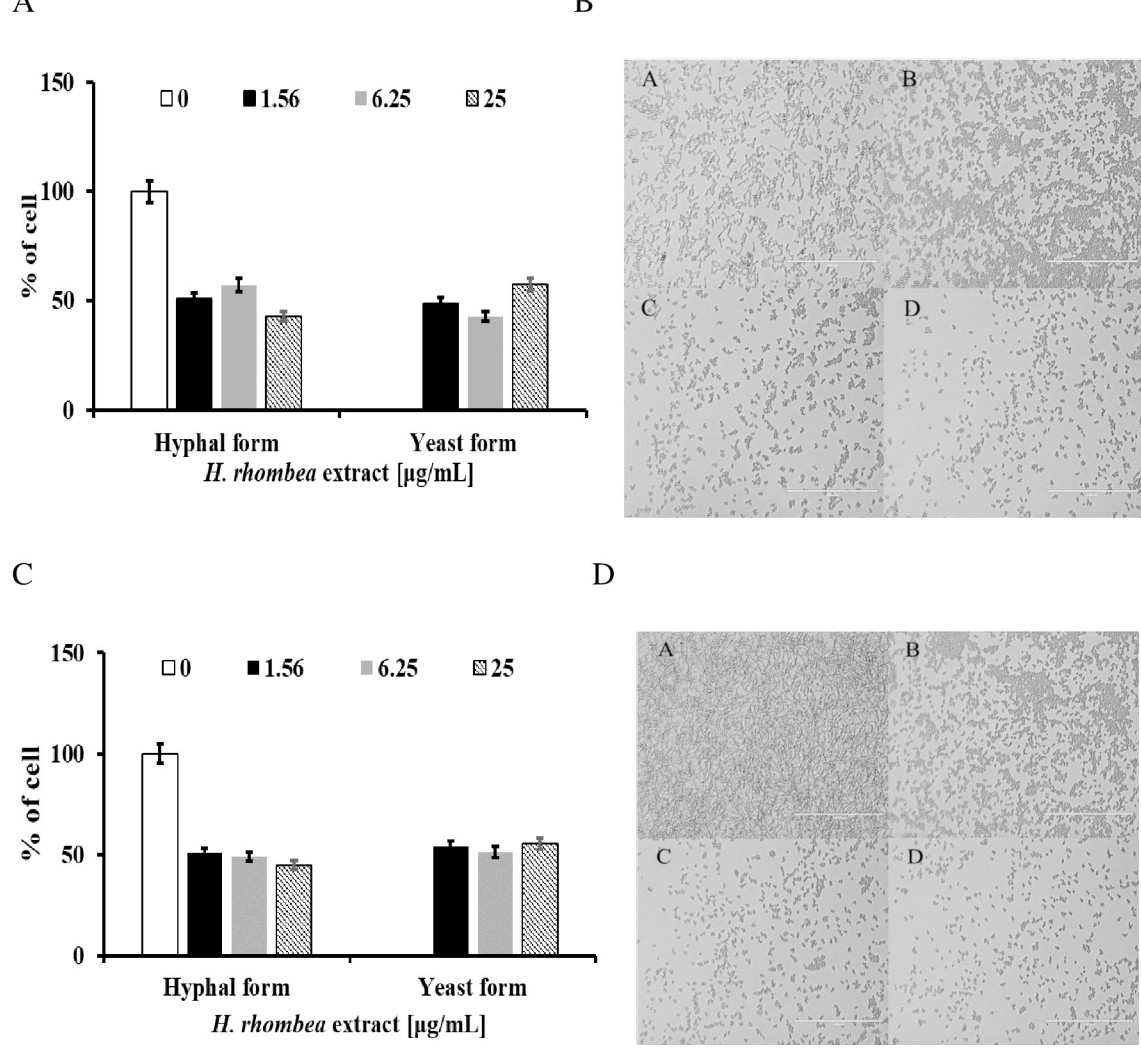

**Fig 3. Inhibition of *C. albicans* dimorphic transition by *H. rhombea* extract in different hyphal-inducing media.** (A) *C. albicans* dimorphic transition was induced using RPMI 1640 and images of *C. albicans* cells (B) were obtained using a microscope. A: *C. albicans* without extract, B: *C. albicans* with 1.56μg/mL of extract treated, C: *C. albicans* with 6.25μg/mL of extract treated, D: *C. albicans* with 25μg/mL of extract treated. (C) *C. albicans* dimorphic transition was induced using 10% FBS YPD medium and images of *C. albicans* cells (D) were obtained using a microscope. A: *C. albicans* without extract, B: *C. albicans* with 1.56μg/mL of extract treated, C: *C. albicans* with 6.25μg/mL of extract treated, D: *C. albicans* with 25μg/mL of extract treated.

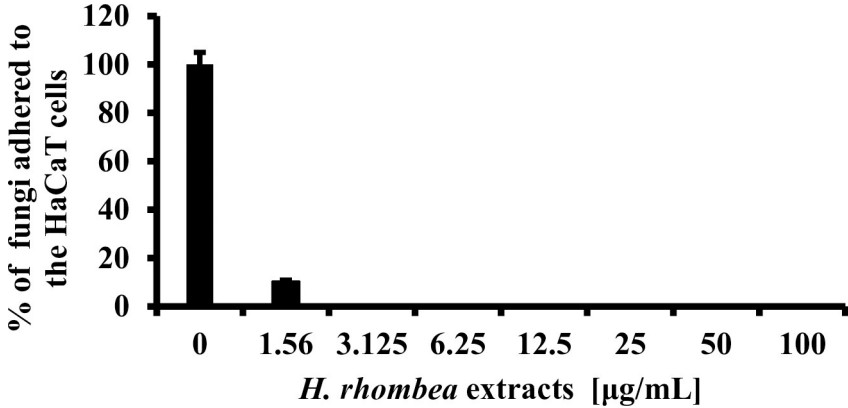

**Fig 4. *H. rhombea* extract inhibited adhesion of C. albicans to HaCaT cells.** *C. albicans* ($1 \times 10^6$ cells/mL) with indicated concentration of *H. rhombea* extract were incubated at 37˚C for 24 h and Candida cells that were remaining on the HaCaT cells were counted.

### *H. rhombea* extract did not inhibit the *Candida* growth

Several antifungal agents inhibit fungal biofilm formation because they can kill the fungi and indirectly decrease biofilm formation. Extract of *H. rhombea* have no effect on the growth of *C. albicans* after 24h of incubation and that suggested the biofilm formation inhibition of extract is not because of the reduced growth of *Candida* (Fig 7).

## Discussion

Many fungi including *Candida* live in and on the human body, but when *Candida* begins to grow uncontrollably, it can cause an infection known as candidiasis. In fact, *Candida* is the most common cause of fungal infections in humans [41]. Antifungal drugs can cause side effects and resistance, and there have been multiple recent reports of resistance including in the emerging problematic organism *Candida auris* [42–44]. Outbreak response is complicated by the limited treatment options and inadequate disinfection strategies. So new approaches with a new target for the anti-fungal agents are required.

In the present study, *H. rhombea* extract was tested for inhibition of *Candida* biofilm formation and showed strong anti-biofilm formation activity against all tested *Candida* species including *C. albicans*, *C. glabrata*, *C. tropicalis*, and *C. parapsilosis* (Fig 1).

Even though the mechanism of *Candida* biofilm formation is diverse depending on the species, *H. rhombea* extract inhibited *Candida* biofilm formation that suggests there should be a common mechanism to induce the biofilm among the *Candida* species, but further studies must be conducted. Moreover, *H. rhombea* increased the antifungal activity of miconazole, magnoflorine, and dioscin (Fig 2), and reduced *Candida* infection (Figs 3 and 4), these results confirmed that biofilm formation is related to the susceptibility of antifungal agents and fungal infection, but further studies should be undertaken. In the case of the *H. rhombea* we tested, no inhibition of growth was observed. However, in the experiments in the reference literature [29], *H. rhombea* extract showed inhibition of growth. The reason might be the differences in the method of extracting plants.

Based on the gene expression analysis, the expression of genes related to biofilm formation, hyphae growth, and cell adhesion was significantly reduced by treatment with *H. rhombea* extract. The biofilm formation is tightly related to adherence to *Candida* and the early step is the attachment mechanism. *C. albicans* biofilm formation is determined by various

A

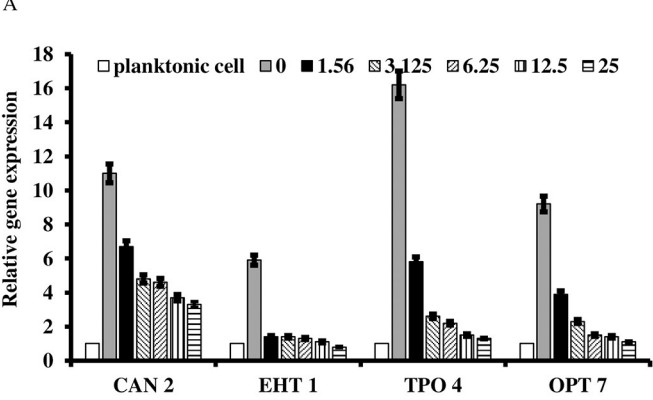

B

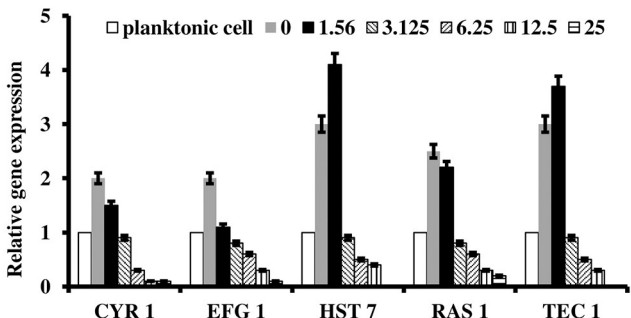

C

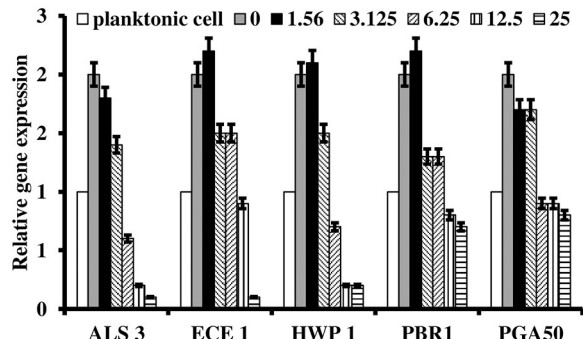

D

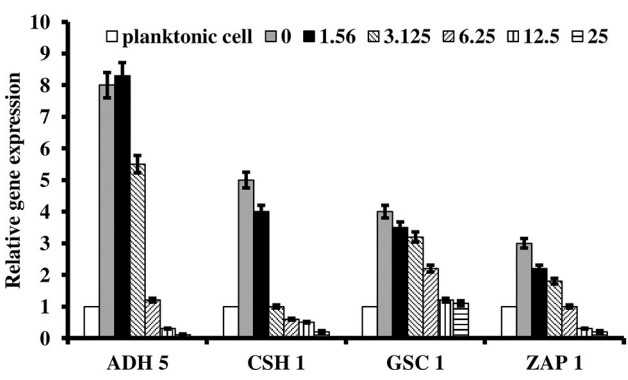

**Fig 5. *H. rhombea* extract inhibited the expression of genes related to biofilm formation and virulence of *C. albicans*.** *H. rhombea* extract reduced the expression of genes related with biofilm formation (A), Ras1-cAMP-Efg1 pathways (B), hyphal-specific (C) and extracellular matrix (D).

transcription factors including *BCR1*, *EFG1*, *TEC1*, *and NDT80* that function as components in several pathways and influence adherence of *Candida*, suggesting that even though biofilm formation was initially tested, adherence or infection factors could be also regulated by treatment of *H. rhombea* extract. Further studies, including the precise target determination of *H. rhombea* extract, effector components, and in vivo testing must be carried out.

A

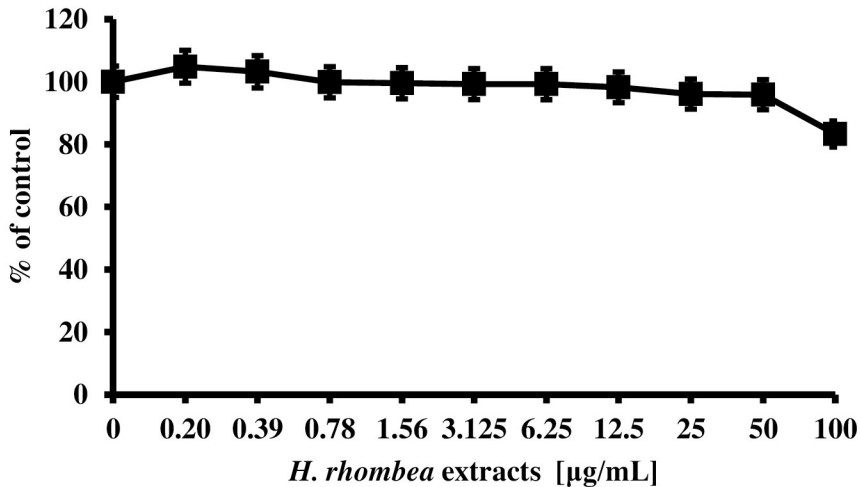

B

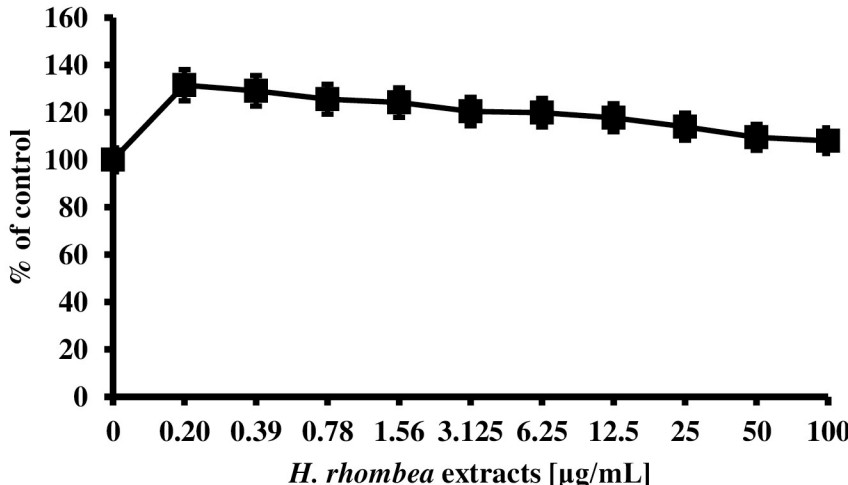

**Fig 6. *H. rhombea* extract did not show any toxicity against the human skin cells.** Cytotoxicity of *H. rhombea* extract against HaCaT cells (A) and THP-1 cells (B). Each cell ($10^4$ per well) was incubated with the indicated concentration of *H. rhombea* extract in 96-well for 24 h and the cell viability was evaluated by MTT assay.

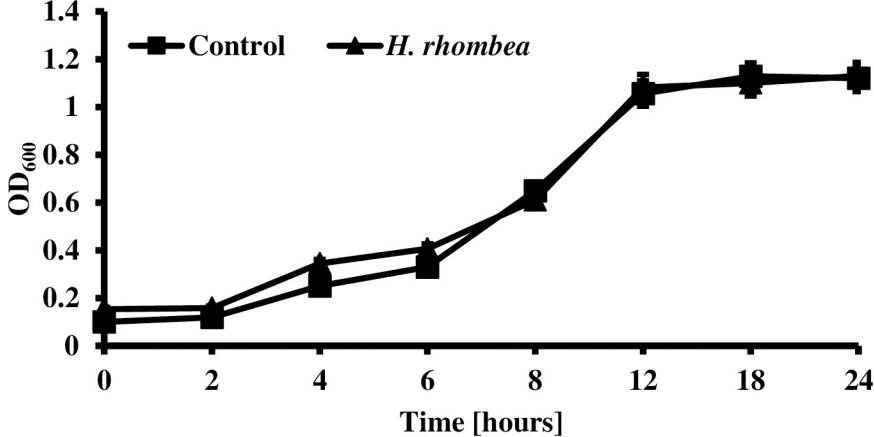

**Fig 7. *H. rhombea* extract did not inhibit the growth of *C. albicans*. *C*. albicans** ($1 \times 10^6$ cells/mL) with 100 μg/mL of *H. rhombea* extract were incubated at 30˚C for 24 h.

In conclusion, *H. rhombea* extract inhibited *C. albicans* biofilm formation, increased the antifungal activity of antibiotics and putative antifungal agents, and decreased fungal adherence to the host cell. Therefore, *H. rhombea* extract could be a good treatment option for biofilm-forming fungal infections.

## Supporting information

**S1 Data.**
(XLSX)

## Author Contributions

**Data curation:** Daseul Kim.

**Formal analysis:** Daseul Kim.

**Funding acquisition:** Ki-young Kim.

**Investigation:** Daseul Kim.

**Supervision:** Ki-young Kim.

**Writing – original draft:** Daseul Kim.

**Writing – review & editing:** Daseul Kim, Ki-young Kim.

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
