## [Decision Letter · Decision Letter 0]

31 Mar 2021

PONE-D-21-06872

Hedera rhombea inhibits the biofilm formation of Candida, thereby increases the susceptibility of antifungal agent, and reduces virulence.

PLOS ONE

Dear Dr. kim,

Thank you for submitting your manuscript to PLOS ONE. After careful consideration, we feel that it has merit but does not fully meet PLOS ONE’s publication criteria as it currently stands. Therefore, we invite you to submit a revised version of the manuscript that addresses the points raised during the review process.

We look forward to receiving your revised manuscript.

Kind regards,

Roy Aziz Khalaf

Academic Editor

PLOS ONE

Additional Editor Comments:

Dear Dr. Kiyoung Kim

Greetings. Two expert reviewers have commented on your manuscript. As you can see there are many issues with the paper before it is suitable for publication, notably in the materials and methods section an how the extract was prepared and with regards to English language. Please make sure to have the manuscript proofread by a native speaker. Please make sure to address all there comments below. In addition, as far as plant extract if a hot water extract from leaves of this plant was performed then this most likely will be acidic. You then concentrated it down and dried it resuspending in DMSO for use. The control is described their control as being no extract. It probably should be DMSO and something should be done about pH change such as buffering (there is no MOPS RPMI), or use an equivalent acid in DMSO as their control.

Thank you

Roy

Journal Requirements:

2. Please include a separate caption for each figure in your manuscript.

https://journals.plos.org/plosone/article?id=10.1371/journal.pone.0167470

In your revision ensure you cite all your sources (including your own works), and quote or rephrase any duplicated text outside the methods section. Further consideration is dependent on these concerns being addressed.

Reviewers' comments:

Reviewer's Responses to Questions

**Comments to the Author**

1. Is the manuscript technically sound, and do the data support the conclusions?

Reviewer #1: Partly

Reviewer #2: Partly

2. Has the statistical analysis been performed appropriately and rigorously? 

Reviewer #1: Yes

Reviewer #2: No

3. Have the authors made all data underlying the findings in their manuscript fully available?

Reviewer #1: Yes

Reviewer #2: Yes

4. Is the manuscript presented in an intelligible fashion and written in standard English?

Reviewer #1: Yes

Reviewer #2: No

5. Review Comments to the Author

Reviewer #1: Language and Format

A native English speaker should be consulted to address language issues. The errors are generally minor, but they are found throughout the paper. For instance, the first sentence of the introduction should read "...with impaired response is able to persist...and often leads to..."

In Candida, gene names are italicized

Title

As noted below, virulence is not tested in this paper.

Abstract

Reference is made to biofilm cellular density, but there are no results regarding this in the paper.

There is reference to bacterial growth.

Introduction

What else is known about extracts from this plant?

Why is there no reference to the following paper, in particular the impact on growth: Choi, H.A.; Cheong, D.E.; Lim, H.D.; Kim, W.H.; Ham, M.H.; Oh, M.H.; Wu, Y.; Shin, H.J.; Kim, G.J. Antimicrobial and anti-biofilm activities of the methanol extracts of medicinal plants against dental pathogens Streptococcus mutans and Candida albicans. J. Microbiol. Biotechnol. 2017, 27, 1242–1248, doi:10.4014/jmb.1701.01026.

Materials and Methods

There are a number of non-standard methods and incomplete descriptions.

In biofilm formation, the inoculum concentration is half standard an no reference is made to inoculum volume. Hypha inducing medium (YPD+serum) is used, but is incubated at yeast temperature (30C). Was the destaining solution transferred to a fresh well for reading the OD?

In the qPCR description, what was the time of the culture with the compound? Was the incubation with the compound done shaking or was this static? How was the cDNA synthesis primed? How were the results normalized?

In the time kill assay, there is a typo (fatal for fetal). Growth is described as being in the YPD and YPD+serum, but the graph has only one base medium.

Results.

It would be helpful if the species name was included at the top of each graph so that the reader didn’t need to keep consulting the figure legend.

In figure 3, it’s not clear if the graphed results are from one replicate or the sum total of all results. There is no n value for the number of cells examined. Why is the cell density so much lower in panel subpanel D of panels B and D than in the other panels? In spite of figure 7, this makes it look like the compound is affecting viability, not just hypha formation. Was the “0” sample without any additive, or with same solvent as the extract?

In figure 4, how many cells were counted?

The times in the graph for Figure 7 do not match those in the methods.

Discussion.

There is a statement that various plant extracts “may not always do their work”. What is that supposed to mean? Figures 3 and 4 do not assess virulence of C. albicans although they examine filamentation and adhesion which are important virulence trait. There are a number of examples of C. albicans strains with adhesion defects which retain high virulence, so this statement needs to be modified.

Reviewer #2: This paper describes the influence of plant extract from Hederea rhombea on Candida albicans. The paper suggest a wide ranging influence of the plant extract making this a paper with potentially high impact.

Their are issues throughout the paper with the English which start with the title. Virulence should not be used as it is not assessed. It should probably read something like "Hedera rhombea inhibits the biofilm formation of Candida and increases the susceptibility of the antifungal agent Miconazole". The issues continue throughout the paper and this really should be improved before further review.

The paper also doesn't address competing data from (https://doi.org/10.4014/jmb.1701.01026 J. Microbiol. Biotechnol. (2017), 27(7), 1242–1248) which showed less influence from Hederea rhombea. This should be referenced and discussed.

The are issues with methods where they are non-standard and not fully described. The controls are either inappropriate or not fully described and this should be remedied. The biofilm data needs error bars and statistical analysis where significant difference is inferred. The protocols for biofilm formation are missing details...like how was the culture prepared prior to preparing cells for biofilm formation. How were the cells prepared for biofilm formation post initial growth. Non-standard film growth conditions are specified but not really justified/explained why 30 not 37 celsius? Why the prolonged incubation with crystal violet. Are they really scoring the actual biofilm and not the destain?. For the dimorphic assay... which RPMI was used how was it made/supplemented. How was it prepared to induce filamentation. The images also need to be better. Without this it is hard to evaluate these experiments. A better label could also be used for graph axis labels than H. rhombea. Perhaps use an acronym for your extract or just put extract.

Below are line by line suggestions that should improve the document. I would also suggest having some very fluent in English review the document.

Abstract suggestions:

change the first line to " Candida is an opportunistic pathogen and common cause of fungal infections worldwide. Anti-fungal use against Candida infections has resulted in the appearance of resistant strains. The limited choice of anti-fungal therapy means alternative strategies are needed to control fungal infectious diseases."

The inhibition of biofilm.....(from line 6) should read something like "Biofilm formation was assessed using the crystal violet assay which showed a dose dependent reduction in the presence of extract"

Genes names should be in italics.

Introduction.

line 1 should read "....pathogen that is responsible for systemic infections in immunocompromised patients....."

line 10 should read "....most antifungal agents, ....."

Line 12. Replace "Thereby" with Therefore.

Line 14 either delete Increasing or change to Increasingly

Line 16 change to "Medicinal plants are used for diverse traditional methods....". end the sentence with "many other diseases"

Line 23 change to "....several fungi..."

Line 24 change to "Particularly, anti-biofilm activity....."

The methods need more detail. Please see above.

Discussion.

Line 3. Change to "Antifungal drugs can cause side effects and resistance and there have been multiple recent reports of resistance including in the emerging problematic organism Candida auris.

At the bottom of the first paragraph. rephrase "those may not always do their work" to be more explicit about what their problem is.

In the third paragraph (and later) ".....further studies must be conducted..." not followed!

Line 18 "... biofilm formation is related to the susceptibility..."

Line 20 "....cell adhesion was significantly reduced by treatment with H. rhombea extract"

on the following/last page

line 3 ".... and in vivo testing must be carried out."

Line 7 is adjuvant really the right word. Perhaps simply say treatment option.

6. PLOS authors have the option to publish the peer review history of their article (what does this mean?). If published, this will include your full peer review and any attached files.

Reviewer #1: No

Reviewer #2: No

---

## [Author Response · Author response to Decision Letter 0]

25 May 2021

Reviewer 1: I have incorporated all your suggestions into my revision. They were helpful. Thank you. Reviewer 2: I have incorporated all your suggestions into my revision. They were helpful. Thank you.

---

## [Decision Letter · Decision Letter 1]

14 Jun 2021

PONE-D-21-06872R1

Hedera rhombea inhibits the biofilm formation of Candida, thereby increases the susceptibility of antifungal agent, and reduces virulence.

PLOS ONE

Dear Dr. kim,

Thank you for submitting your manuscript to PLOS ONE. After careful consideration, we feel that it has merit but does not fully meet PLOS ONE’s publication criteria as it currently stands. Therefore, we invite you to submit a revised version of the manuscript that addresses the points raised during the review process.

Dear Sir

The reviewers have assessed your manuscript and both agree that even though it is vastly improved it still needs some work. Please modify according to the reviewer comments, especially those of reviewer two.

Thank you

We look forward to receiving your revised manuscript.

Kind regards,

Roy Aziz Khalaf

Academic Editor

PLOS ONE

Journal Requirements:

Additional Editor Comments (if provided):

Dear Sir

The reviewers have assessed your manuscript and both agree that even though it is vastly improved it still needs some work. Please modify according to the reviewer comments, especially those of reviewer two.

Thank you

Reviewers' comments:

Reviewer's Responses to Questions

**Comments to the Author**

1. If the authors have adequately addressed your comments raised in a previous round of review and you feel that this manuscript is now acceptable for publication, you may indicate that here to bypass the “Comments to the Author” section, enter your conflict of interest statement in the “Confidential to Editor” section, and submit your "Accept" recommendation.

Reviewer #1: (No Response)

Reviewer #2: (No Response)

2. Is the manuscript technically sound, and do the data support the conclusions?

Reviewer #1: Partly

Reviewer #2: Partly

3. Has the statistical analysis been performed appropriately and rigorously? 

Reviewer #1: Yes

Reviewer #2: No

4. Have the authors made all data underlying the findings in their manuscript fully available?

Reviewer #1: Yes

Reviewer #2: Yes

5. Is the manuscript presented in an intelligible fashion and written in standard English?

Reviewer #1: No

Reviewer #2: Yes

6. Review Comments to the Author

Reviewer #1: Methods and Results

Please make sure that the temperatures in the methods match the temperatures in the figure legends.

As a formatting issue, I’m not sure why the figure legends are incorporated into the results section.

I have concerns with the results reported in figure 3 (inhibition of filamentation). The following description is difficult to understand “This method, which was necessary because the hyphae-specific identification cells could not be quantified, also resulted in an under expression of the number of mycelial cells in the population. The reported percentage of hyphae is normalized to the percentage of probability hyphae when a numerator is not added. Since the number of cells observed in D is small, it seems to be the part where the number of observed cells is small.”

What does “the percentage of probability hyphae” mean?

The table in the supplementatal information indicates that the percentage graphed is the average number from a particular treatment as a percentage of the number in the untreated control. Presumably these numbers are the number of hyphal cells. I don’t see the rationale for expressing this information as a percentage of the control. Surely graphing the percentage of hyphal cells out of the total number of cells counted in a particular treatment would be clearer.

Numerous language issues remain.

Throughout it should be susceptibility to an antifungal, not susceptibility of an antifungal.

Line 52 should have “pathogenesis” instead of pathogenic.

Line 57 The following does not make sense, although I have an idea of what the second part is trying to say “The biofilm matrix acts as an intestinal for sessile cells, avert the inlet of the majority used antifungal agents, therefore give drug resistance.”

Line 65 should read “Traditional medicine is proposed to have lower adverse reactions compare with typical medicines.”

Line 92 What does “Controls without test compounds were served” mean?

Line 95 something like “for 24h, non-adherent cells were removed by washing with PBS and then 100ul of 1% aqueous crystal violet was applied for 30 minutes” might be better

Line 112 (and others) should be fetal, not fatal.

Line 116: the term bud would be best, not “sprout cell” and this whole description could be simplified.

Line 133 should read “non-adherent” for “none adhere”

Line 158. This might be better described as a growth inhibition assay, as that is what is being assessed, not killing.

Line 187 should read “Extract was applied”

Line 189 should read “increased the susceptibility to miconazole”

Line 190 I think this line should read “The extract also increased the susceptibility to plant-derived antifungal candidates including”

Fig 2 Legend should read increased the susceptibility of C. albicans to miconazole etc.

Line 205: what is meant by “serious infection effect”?

Line 286: perhaps “further studies must be conducted.”

Line 293: gene names must be italicized

Line 294: perhaps “that influence adhesion” in place of “and is also participated to the adherence”

Line 295 “could” would be better than “should”

Line 296 should read “Further studies, including…”

In figure 5 Planktonic has been misspelled

Reviewer #2: This revised manuscript is much improved, I thank the authors for their hard work.

I still have some remaining concerns and a few language corrections.

My major concern relates to their plant leaf extract and the use of RPMI. Many plant leaf extracts are acidic.

RPMI as used in the candida field is typically heavily buffered with MOPS. It is not clear from the description of the methods that RPMI was buffered. Candida filamentation and biofilm formation are heavily influenced by pH. Please clarify the situation with the RPMI.

My second concern with the revision is that the discussion still does not address competing (and conflicting data) (https://doi.org/10.4014/jmb.1701.01026 J. Microbiol. Biotechnol. (2017), 27(7), 1242–1248) which showed less influence from Hederea rhombea. This should be referenced and discussed.

The controls still aren't clear. Presumably these are just DMSO. But this needs to be clearly stated when it is used. There is a comment that "controls without test compounds were served" but it should clearly state somewhere is the 0/control used is DMSO

The title is misleading and should really read something like "Hedera rhombea inhibits the biofilm formation of Candida and increases the susceptibility to antifungal agents"

Below are minor language changes to improve the manuscript.

Abstract/Page 8

line33 should read "Treatment with 6.25....." and "...increased susceptibility to miconazole..."

line39 should read "...analysis was done and the extract was found to dose dependently reduce the expression....

Page 9

line 44 should read " biofilm formation is good screen for developing.....

Introduction Page 10

line 50 should read "....persist inside the host and can be aided by drug resistance traits which often lead to failure of....."

line 52 should read "...species pathogenicity is their..."

line 56 should read "...is a finely controlled process that involves attachment to surface and embedment in the exopolymer...."

line 57 should read "the biofilm matrix acts as a haven for sessile cells..."

line 58 should read "....averting the inlet of the majority of anitfungal agents, thereby resulting in resistance"

line 59 I think the authors mean to say "adherent C. albicans cells without specific drug resistant gene expression..."

line 67 should read "...provenance of replacement remedies...."

line 73 should read "Interestingly, the activity of the extract also increased susceptibility to antibiotics."

Methods

line 92 should probably read "DMSO controls without test compounds were used"

???line 93 ...was this RPMI MOPS buffered????

line 112... I think autocorrect likely altered this as fatal bovine serum is back/still there. It should be fetal or foetal. This error recurs...see line 130

Results

line 185 should read "H. rhombea extract increased the efficacy of antifungal agents...."

If you want to use susceptibility then the extract would increase the susceptibility of C. albicans to antifungals.

Discussion

line 285-286. should read "....but further studies should be undertaken."

7. PLOS authors have the option to publish the peer review history of their article (what does this mean?). If published, this will include your full peer review and any attached files.

Reviewer #1: No

Reviewer #2: No

---

## [Author Response · Author response to Decision Letter 1]

30 Jul 2021

Hedera rhombea inhibits the biofilm formation of Candida, thereby increases the susceptibility to antifungal agent, and reduces infection.

We thank the editors and the reviewers for their thoughtful and helpful comments. We have addressed, in a point-by-point manner, all the suggestions and queries from the journal and the reviewer and marked with red in manuscript. The input from the reviewers has allowed us to improve the clarity and quality of our paper. We have included below our point-by-point response to the reviewers’ comments and have included these additions and alterations to the revised manuscript.

Reviewers' comments:

Reviewer's Responses to Questions

Comments to the Author

1. If the authors have adequately addressed your comments raised in a previous round of review and you feel that this manuscript is now acceptable for publication, you may indicate that here to bypass the “Comments to the Author” section, enter your conflict of interest statement in the “Confidential to Editor” section, and submit your "Accept" recommendation.

Reviewer #1: (No Response)

Reviewer #2: (No Response)

2. Is the manuscript technically sound, and do the data support the conclusions?

Reviewer #1: Partly

Reviewer #2: Partly

3. Has the statistical analysis been performed appropriately and rigorously?

Reviewer #1: Yes

Reviewer #2: No

4. Have the authors made all data underlying the findings in their manuscript fully available?

Reviewer #1: Yes

Reviewer #2: Yes

5. Is the manuscript presented in an intelligible fashion and written in standard English?

Reviewer #1: No

Reviewer #2: Yes

6. Review Comments to the Author

Reviewer #1: Methods and Results

Please make sure that the temperatures in the methods match the temperatures in the figure legends. As a formatting issue, I’m not sure why the figure legends are incorporated into the results section.

Sorry to make confusing the figure legends. We fixed the temperature.

Fig 1. Inhibition of Canddia spp. biofilm formation by H. rhombea extract

The biofilm formation of (A) C. albicans, (B) C. tropicalis var. tropicalis, (C) C. parapsilosis var. parapsilosis, (D) C. glabrata, (E) C. tropicalis, and (F) C. parapsilosis was induced in YPD media supplemented with 10% fatal bovine serum with the indicated concentrations of H. rhombea extract at 37 ℃ for 24h.

Fig 2. H. rhombea extracts increased the susceptibility of C. albicans to miconazole etc.

H. rhombea extract (6.25μg/mL) increased the susceptibility of miconazole (3.125μg/mL) (A), magnoflorine (3.125μg/mL) (B) and dioscin (3.125μg/mL) (c) against C. albicans. The biofilm formation was induced for 24 h by growing of C. albicans in YPD media supplemented with 10% fetal bovine serum. MCZ: Miconazole, MF: Magnoflorine, Do: Dioscin, HR: H. rhombea extract.

Fig. 3 Inhibition of C. albicans dimorphic transition by H. rhombea extract in different hyphal-inducing media.

(A) C. albicans dimorphic transition was induced using RPMI 1640 and images of C. albicans cells (B) were obtained using a microscope. A: C. albicans without extract, B: C. albicans with 1.56μg/mL of extract treated, C: C. albicans with 6.25μg/mL of extract treated, D: C. albicans with 25μg/mL of extract treated.

(C) C. albicans dimorphic transition was induced using 10% FBS YPD medium and images of C. albicans cells (D) were obtained using a microscope. A: C. albicans without extract, B: C. albicans with 1.56μg/mL of extract treated, C: C. albicans with 6.25μg/mL of extract treated, D: C. albicans with 25μg/mL of extract treated.

Fig. 4 H. rhombea extract inhibited adhesion of C. albicans to HaCaT cells.

C. albicans (1 × 106 cells/mL) with indicated concentration of H. rhombea extract were incubated at 37°C for 24 h and Candida cells that were remaining on the HaCaT cells were counted.

Fig. 5 H. rhombea extract inhibited the expression of genes related to biofilm formation and virulence of C. albicans.

H. rhombea extract reduced the expression of genes related with biofilm formation (A), Ras1-cAMP-Efg1 pathways (B), hyphal-specific (C) and extracellular matrix (D).

Total RNA was extracted from C. albicans treated with the indicated concentration of H. rhombea extract using RNA extraction kit, converted to cDNA, and analyzed by qPCR with the respective primers. 

I have concerns with the results reported in figure 3 (inhibition of filamentation). The following description is difficult to understand “This method, which was necessary because the hyphae-specific identification cells could not be quantified, also resulted in an under expression of the number of mycelial cells in the population. The reported percentage of hyphae is normalized to the percentage of probability hyphae when a numerator is not added. Since the number of cells observed in D is small, it seems to be the part where the number of observed cells is small.”

: Thanks for your suggestion. We have modified the experimental method for the problematic part.

C. albicans were grown overnight in YPD medium. 1 × 106 cells/mL of Candida with or without extracts were incubated in RPMI 1640 medium and YPD media supplemented with 10% fetal bovine serum medium to induce dimorphic transition at 37°C for 4 h. Inhibition quantification of the yeast-to-hyphal-form transition was accomplished by counting the number of hyphae cells in the population as previously described [26, 32, 33, 34, 35, 36, 37, 38, 39, 40]. More than 1,000 cells were counted for each well in duplicate, and all assays were repeated five times. Representative results of images were obtained using a fluorescence microscope (EVOS® FL, ThermoFisher Scientific, Waltham, MA, USA). The experiments were performed in triplicate. 

What does “the percentage of probability hyphae” mean?

: Thanks for your suggestion. We have modified the experimental method for the problematic part.

C. albicans were grown overnight in YPD medium. 1 × 106 cells/mL of Candida with or without extracts were incubated in RPMI 1640 medium and YPD media supplemented with 10% fetal bovine serum medium to induce dimorphic transition at 37°C for 4 h. Inhibition quantification of the yeast-to-hyphal-form transition was accomplished by counting the number of hyphae cells in the population as previously described [26, 32, 33, 34, 35, 36, 37, 38, 39, 40]. More than 1,000 cells were counted for each well in duplicate, and all assays were repeated five times. Representative results of images were obtained using a fluorescence microscope (EVOS® FL, ThermoFisher Scientific, Waltham, MA, USA). The experiments were performed in triplicate. 

The table in the supplementatal information indicates that the percentage graphed is the average number from a particular treatment as a percentage of the number in the untreated control. Presumably these numbers are the number of hyphal cells. I don’t see the rationale for expressing this information as a percentage of the control. Surely graphing the percentage of hyphal cells out of the total number of cells counted in a particular treatment would be clearer.

Numerous language issues remain. Throughout it should be susceptibility to an antifungal, not susceptibility of an antifungal.

Fig. 3

(A)

(C)

Line 52 should have “pathogenesis” instead of pathogenic.

: Thanks for your suggestion. Based on your advice, we have changed word.

One of the features of Candida species pathogenesis is their ability to form biofilms, and nosocomial infections are often related to the ability to produce biofilm on mucosal surfaces and implanted medical devices [2, 3, 4, 5].

Line 57 The following does not make sense, although I have an idea of what the second part is trying to say “The biofilm matrix acts as an intestinal for sessile cells, avert the inlet of the majority used antifungal agents, therefore give drug resistance.”

: Thank you very much. We made the corrections as you mentioned.

The biofilm matrix highly acts to structure microbial communities include sessile cells that are frequently much more resistant to antifungal agents 

Line 65 should read “Traditional medicine is proposed to have lower adverse reactions compare with typical medicines.”

: Thanks for your constructive suggestions. We have made corrections in that regard based on your advice.

Herbal medicines usually have fewer side effects compared to over-the-counter medicines. 

Line 92 What does “Controls without test compounds were served” mean?

: Thank you very much. Based on your advice, we have changed word.

Wells without test compounds served as controls (DMSO concentration of 0.1%).

Line 95 something like “for 24h, non-adherent cells were removed by washing with PBS and then 100ul of 1% aqueous crystal violet was applied for 30 minutes” might be better

: Thank you for your suggestion. Based on your advice, we have added the manuscript modified portion shown on line 95.

After incubation at 37°C for 24h, non-adherent cells were removed by washing with PBS and then 100ul of 1% aqueous crystal violet was applied for 30 minutes.

Line 112 (and others) should be fetal, not fatal.

: Thank you very much. Based on your advice, we have changed word.

C. albicans were grown overnight in YPD medium. 1 × 106 cells/mL of Candida with or without extracts were incubated in RPMI 1640 medium and YPD media supplemented with 10% fetal bovine serum medium to induce dimorphic transition at 37°C for 4 h.

Line 116: the term bud would be best, not “sprout cell” and this whole description could be simplified.

: Thanks for your constructive suggestions. We have modified the experimental method for the problematic part.

C. albicans were grown overnight in YPD medium. 1 × 106 cells/mL of Candida with or without extracts were incubated in RPMI 1640 medium and YPD media supplemented with 10% fetal bovine serum medium to induce dimorphic transition at 37°C for 4 h. Inhibition quantification of the yeast-to-hyphal-form transition was accomplished by counting the number of hyphae cells in the population as previously described [26, 32, 33, 34, 35, 36, 37, 38, 39, 40]. More than 1,000 cells were counted for each well in duplicate, and all assays were repeated five times. Representative results of images were obtained using a fluorescence microscope (EVOS® FL, ThermoFisher Scientific, Waltham, MA, USA). The experiments were performed in triplicate. 

Line 133 should read “non-adherent” for “none adhere”

: Thank you very much. we have added the manuscript modified portion shown on line 133.

DMEM were drained and then plates were cautiously washed three times with PBS to remove none adhere cell.

Line 158. This might be better described as a growth inhibition assay, as that is what is being assessed, not killing.

: Thanks for your suggestion. Modified according to your advice. 

Growth inhibition assay for C. albicans

Line 187 should read “Extract was applied”

: Thank you very much. We have made corrections in that regard based on your advice.

H. rhombea extract was applied increased the susceptibility of antifungal agents against C. albicans. 

Line 189 should read “increased the susceptibility to miconazole”

: Thank you for your careful work. We have added changed of the word.

Treatment of 3.125μg/mL of extract increased the susceptibility to miconazole with 99% of fungal growth inhibition from 38% inhibition by miconazole treatment.

Line 190 I think this line should read “The extract also increased the susceptibility to plant-derived antifungal candidates including”

: Thank you for your professional advice. According to the guide for author, we carefully revised changed the word.

The extract also increased the susceptibility to plant-derived antifungal candidates including magnoflorine (99% growth inhibition by 6.25μg/mL of extract compared with 29% growth inhibition by 3.125ug/ml of only magnoflorine treatment) and dioscin (99% growth inhibition by 6.25μg/mL of extract compared with 39% growth inhibition by 3.125ug/ml of only dioscin treatment) (Fig 2A, 2B and 2C).

Fig 2 Legend should read increased the susceptibility of C. albicans to miconazole etc.

: Thank you for your valuable suggestions. We have modified

Fig 2. H. rhombea extract increased the susceptibility of C. albicans to miconazole etc.

Line 205: what is meant by “serious infection effect”?

: Thank you for your careful work. We have added changed of the word.

The yeast-to-hyphae conversion is an important virulence-mediated property of C. albicans. The formation of hyphae aids the subsequent invasive growth of C. albicans to penetrate host tissues and lead to the establishment of systemic infection [7]. Extract tested whether the dimorphic transition was influenced and found 1.56μg/mL of the extract significantly inhibited hyphae formation in RPMI 1640 or a 10% FBS YPD medium, and extract with higher than 6.25μg/mL completely blocked the hyphae formation (Fig 3A, 3B, 3C and 3D). 

Line 286: perhaps “further studies must be conducted.”

: Thank you for your constructive suggestion. As you suggested.

H. rhombea extract inhibited Candida biofilm formation that suggests there should be a common mechanism to induce the biofilm among the Candida species, but further studies must be conducted.

Line 293: gene names must be italicized

: Thanks for your constructive suggestions. As you suggest, I have corrected that part.

Candida species biofilm formation is determined by various transcription factors including BCR1, EFG1, TEC1, and NDT80 that function as a component part in several pathways is also participated to the adherence of Candida suggested that even though biofilm formation is initially tested, adherence or infection factors could be also regulated by treatment of H. rhombea extract. 

Line 294: perhaps “that influence adhesion” in place of “and is also participated to the adherence”

: Thank you for your careful reading of our manuscript. According to your comments, we have carefully changed this part.

Candida species biofilm formation is determined by various transcription factors including BCR1, EFG1, TEC1, and NDT80 that function as a component part in several pathways and is also participated to the adherence of Candida suggested that even though biofilm formation is initially tested, adherence or infection factors could be also regulated by treatment of H. rhombea extract. 

Line 295 “could” would be better than “should”

: Thank you for your valuable comments. According to your suggestion.

Candida species biofilm formation is determined by various transcription factors including BCR1, EFG1, TEC1, and NDT80 that function as a component part in several pathways is also participated to the adherence of Candida suggested that even though biofilm formation is initially tested, adherence or infection factors could be also regulated by treatment of H. rhombea extract. 

Line 296 should read “Further studies, including…”

: Thanks for your constructive suggestion and we have made the modification. 

Further studies, including the precise target determination of H. rhombea extract, effector components, and in vivo testing must be carried out.

In figure 5 Planktonic has been misspelled

: Thank you for your careful review. The part you mentioned has been removed.

Reviewer #2: This revised manuscript is much improved, I thank the authors for their hard work. I still have some remaining concerns and a few language corrections.

My major concern relates to their plant leaf extract and the use of RPMI. Many plant leaf extracts are acidic.

RPMI as used in the candida field is typically heavily buffered with MOPS. It is not clear from the description of the methods that RPMI was buffered. Candida filamentation and biofilm formation are heavily influenced by pH. Please clarify the situation with the RPMI.

: Thank you for your careful reading of our manuscript. We conducted the experiment with reference to these three reference documents.

Yang L, Liu X, Zhuang X, Feng X, Zhong L, Ma T. Antifungal Effects of Saponin Extract from Rhizomes of Dioscorea panthaica Prain et Burk against Candida albicans. Evid Based Complement Alternat Med 2018; 2018:6095307. doi: 10.1155/2018/6095307 PMID: 29853962

Zhang LL, Lin H, Liu W, Dai B, Yan L, Cao YB, et al. Antifungal Activity of the Ethanol Extract from Flos Rosae Chinensis with Activity against Fluconazole-Resistant Clinical Candida. Evid Based Complement Alternat Med. 2017; 2017:4780746. doi: 10.1155/2017/4780746 PMID: 28303159

Bonifácio BV, Vila TVM, Masiero IF, Silva PBD, Silva ICD, Lopes ÉDO, et al. Antifungal Activity of a Hydroethanolic Extract From Astronium urundeuva Leaves Against Candida albicans and Candida glabrata. Front Microbiol 2019; 10: 2642. doi: 10.3389/fmicb.2019.02642 PMID: 31803166

My second concern with the revision is that the discussion still does not address competing (and conflicting data) (https://doi.org/10.4014/jmb.1701.01026 J. Microbiol. Biotechnol. (2017), 27(7), 1242–1248) which showed less influence from Hederea rhombea. This should be referenced and discussed.

: Thank you very much. We carefully checked the references again to make sure was correct.

In the case of the H. rhombea we tested, no inhibition of growth was observed. However, in the experiments in the reference literature [29], it was confirmed that the result of H. rhombea inhibiting growth. The reason might be the differences in the method of extracting plants.

The controls still aren't clear. Presumably these are just DMSO. But this needs to be clearly stated when it is used. There is a comment that "controls without test compounds were served" but it should clearly state somewhere is the 0/control used is DMSO

: Thanks for your careful work. As you mentioned, I have edited and written that part.

Wells without test compounds served as controls (DMSO concentration of 0.1%).

The title is misleading and should really read something like "Hedera rhombea inhibits the biofilm formation of Candida and increases the susceptibility to antifungal agents"

Below are minor language changes to improve the manuscript.

: Thank you so much. We have checked the whole manuscript carefully according to your advice to improve the readability and accuracy.

Hedera rhombea inhibits the biofilm formation of Candida and increases the susceptibility to antifungal agents

Abstract/Page 8

line33 should read "Treatment with 6.25....." and "...increased susceptibility to miconazole..."

: Thank you for your constructive suggestions. According to the guide for author, we carefully revised the word.

Treatment with 6.25μg/mL of extract increased susceptibility to miconazole from 32% decreasing of fungal growth to 98.8% of that based on the fungal growth assay.

line39 should read "...analysis was done and the extract was found to dose dependently reduce the expression....

: Thank you very much. We have checked and corrected the points you mentioned.

To know the molecular mechanisms of biofilm formation inhibition by extract, qRT-PCR analysis was done, and the extract was found to dose dependently reduce the expression of hyphal-associated genes (ALS3, ECE1, HWP1, PGA50, and PBR1), extracellular matrix genes (GSC1, ZAP1, ADH5, and CSH1), Ras1-cAMP-PKA pathway genes (CYR1, EFG1, and RAS1), Cph2-Tec1 pathway gene (TEC1) and MAP kinases pathway gene (HST7).

Page 9

line 44 should read " biofilm formation is good screen for developing.....

: Thank you for pointing this out. We have corrected the word you pointed out.

In this study, Hedera rhombea extract showed inhibition of fungal biofilm formation, activation of antifungal susceptibility, and reduction of infection. These results suggest that fungal biofilm formation is good screen for developing the antifungal adjuvant and Hedera rhombea extract should be a good candidate against biofilm-related fungal infection.

Introduction Page 10

line 50 should read "....persist inside the host and can be aided by drug resistance traits which often lead to failure of....."

: Thank you for your constructive suggestion. As you suggested, we changed word.

C. albicans can persist inside the host and can be aided by drug resistance traits which often lead to failure of therapeutic strategies [1].

line 52 should read "...species pathogenicity is their..."

: Thank you for your constructive suggestion. We have edited that part.

line 56 should read "...is a finely controlled process that involves attachment to surface and embedment in the exopolymer...."

: Thank you very much. We have made the modification.

The formation of biofilms involves multiple interconnected signaling pathways [6, 7, 8, 9, 10, 11, 12, 13, 14], and is a finely controlled process that involves attachment to surface and embedment in the exopolymer extracellular matrix [15, 16, 17, 18].

line 57 should read "the biofilm matrix acts as a haven for sessile cells..."

: Thank you for your constructive suggestion. We have edited that part.

line 58 should read "....averting the inlet of the majority of anitfungal agents, thereby resulting in resistance"

: Thank you for your constructive suggestion. We have edited that part.

line 59 I think the authors mean to say "adherent C. albicans cells without specific drug resistant gene expression..."

: Thank you for your valuable suggestions. We have modified

In fact, adherent C. albicans cells without specific drug resistant gene expression are up to 1,000 times more resistant to common antifungal agents than planktonic cells [19].

line 67 should read "...provenance of replacement remedies...."

: Thanks for your suggestion. Modified according to your advice.

Accordingly, medicinal plants should be a new provenance of replacement remedies to treat Candida infectious diseases [24, 25, 26].

line 73 should read "Interestingly, the activity of the extract also increased susceptibility to antibiotics."

: Thank you very much. We made the corrections as you mentioned.

Interestingly, the activity of the extract also increased susceptibility to antibiotics and reducing fungal infection.

Methods

line 92 should probably read "DMSO controls without test compounds were used"

: Thank you for your valuable comments. According to your suggestion.

DMSO controls without test compounds were used.

line 93 ...was this RPMI MOPS buffered????

: Thank you for your careful reading of our manuscript. We conducted the experiment with reference to these three reference documents.

Yang L, Liu X, Zhuang X, Feng X, Zhong L, Ma T. Antifungal Effects of Saponin Extract from Rhizomes of Dioscorea panthaica Prain et Burk against Candida albicans. Evid Based Complement Alternat Med 2018; 2018:6095307. doi: 10.1155/2018/6095307 PMID: 29853962

Zhang LL, Lin H, Liu W, Dai B, Yan L, Cao YB, et al. Antifungal Activity of the Ethanol Extract from Flos Rosae Chinensis with Activity against Fluconazole-Resistant Clinical Candida. Evid Based Complement Alternat Med. 2017; 2017:4780746. doi: 10.1155/2017/4780746 PMID: 28303159

Bonifácio BV, Vila TVM, Masiero IF, Silva PBD, Silva ICD, Lopes ÉDO, et al. Antifungal Activity of a Hydroethanolic Extract From Astronium urundeuva Leaves Against Candida albicans and Candida glabrata. Front Microbiol 2019; 10: 2642. doi: 10.3389/fmicb.2019.02642 PMID: 31803166

line 112... I think autocorrect likely altered this as fatal bovine serum is back/still there. It should be fetal or foetal. This error recurs...see line 130

: Thank you for pointing this out. We have corrected the word you pointed out.

C. albicans were grown overnight in YPD medium. 1 × 106 cells/mL of Candida with or without extracts were incubated in RPMI 1640 medium and YPD media supplemented with 10% fetal bovine serum medium to induce dimorphic transition at 37°C for 4 h.

A human epithelial keratinocyte HaCaT cells were maintained in DMEM supplemented with 10% fetal bovine serum at 37°C in a humidified atmosphere of 5% CO2.

Results

line 185 should read "H. rhombea extract increased the efficacy of antifungal agents...."

If you want to use susceptibility then the extract would increase the susceptibility of C. albicans to antifungals.

: Thank you for your careful reading of our manuscript. we have carefully changed this part.

Discussion

line 285-286. should read "....but further studies should be undertaken."

: Thank you for your valuable suggestions. We have modified the word.

Moreover, H. rhombea increased the antifungal activity of miconazole, magnoflorine, and dioscin (Fig 2), and reduced Candida infection (Fig 3 and 4), these results confirmed that biofilm formation is related to the susceptibility of antifungal agents and fungal infection, but further studies should be undertaken.

---

## [Decision Letter · Decision Letter 2]

18 Aug 2021

PONE-D-21-06872R2

Hedera rhombea inhibits the biofilm formation of Candida, thereby increases the susceptibility to antifungal agent, and reduces infection.

PLOS ONE

Dear Dr. kim,

Thank you for submitting your manuscript to PLOS ONE. After careful consideration, we feel that it has merit but does not fully meet PLOS ONE’s publication criteria as it currently stands. Therefore, we invite you to submit a revised version of the manuscript that addresses the points raised during the review process.

Dear Dr. Kim.

Greetings. The paper is much improved. However minor revisions are still required especially the issue of the RPMI media highlighted by reviewer 2 before the manuscript can be officially accepted. Please address the reviewer comments below.

Thank you

We look forward to receiving your revised manuscript.

Kind regards,

Roy Aziz Khalaf

Academic Editor

PLOS ONE

Journal Requirements:

Additional Editor Comments (if provided):

Dear Dr. Kim.

Greetings. The paper is much improved. However minor revisions are still required especially the issue of the RPMI media highlighted by reviewer 2 before the manuscript can be officially accepted. Please address the reviewer comments below.

Thank you

Reviewers' comments:

Reviewer's Responses to Questions

**Comments to the Author**

1. If the authors have adequately addressed your comments raised in a previous round of review and you feel that this manuscript is now acceptable for publication, you may indicate that here to bypass the “Comments to the Author” section, enter your conflict of interest statement in the “Confidential to Editor” section, and submit your "Accept" recommendation.

Reviewer #1: (No Response)

Reviewer #2: (No Response)

2. Is the manuscript technically sound, and do the data support the conclusions?

Reviewer #1: Yes

Reviewer #2: Yes

3. Has the statistical analysis been performed appropriately and rigorously? 

Reviewer #1: Yes

Reviewer #2: Yes

4. Have the authors made all data underlying the findings in their manuscript fully available?

Reviewer #1: Yes

Reviewer #2: Yes

5. Is the manuscript presented in an intelligible fashion and written in standard English?

Reviewer #1: Yes

Reviewer #2: Yes

6. Review Comments to the Author

Reviewer #1: I appreciate the changes made by the authors and I have only a handful of minor edits remaining.

Line 57 should read “The biofilm matrix acts to structure microbial communities and includes sessile cells that are frequently much more resistant to antifungal agents.”

Line 74 I think “and reducing fungal infection” needs to be removed since virulence wasn’t assessed in the manuscript.

Line 125 should read “nonadherent”

Line 179 remove “was applied”

Line 196 should just read “virulence” not “virulence-mediated”

Line 290 I think this should read ”Candida albicans biofilm formation is determined by various transcription factors including BCR1, EFG1, TEC1, and NDT80 that function as components in several pathways and also influence adherence of Candida, suggesting that even though biofilm formation was initially tested, adherence or infection factors could be also regulated by treatment of H. rhombea extract.”

Figure 1 Legend has one last “fatal” instead of “fetal”

Figure 5 be sure that in the figure the label is “planktonic”.

Reviewer #2: This revised manuscript is much improved, I thank the authors for their

hard work. My main remaining concern and is with regards to the RPMI based experiments. The authors provided references to respond to this question. The frontiers paper that was referenced clearly describes the RPMI they used as supplemented with L-glutamine and buffered with MOPS to pH 7. (see extract 1 below..copied from the article).

The other two papers reference the CLSI standard and then they describe RPMI without sodium barcarbonate at pH7. The CLSI standard recommends the use of RPMI without sodium bicarbonate and with glutamine...but it also says to use a buffer and recommends mops (see second extract below).

If this is accurate for the experiments described here then a similar line should be added to methods. If not perhaps the authors could include a line about the pH of their extract. Or comment on the color of the media as the phenol red is in RPMI as a pH indicator.

Alternatively RPMI can use a sodium bicarbonate/Co2 buffer system....but that is usually for cell culture.

[extract 1] taken from Front. Microbiol., 15 November 2019 | https://doi.org/10.3389/fmicb.2019.02642 "A round-bottom 96-well microplate (Corning Inc., Corning, NY, United States) was filled with RPMI medium supplemented with L-glutamine (Cellgro; Corning, United States) and buffered with 165 mM morpholinepropanesulfonic acid (MOPS) (Thermo Fisher Scientific Inc., United States) at pH 7.0 "

[extract 2] taken from CLSI M27 standard for antifungal testing: "Media should be buffered to a pH of 7.0 ± 0.1 at 25 °C. A buffer should be selected that does not antagonize antifungal agents. Tris buffer is unsatisfactory, because it antagonizes the activity of flucytosine. Zwitterion buffers are preferable to buffers that readily traverse the cell membrane, such as phosphate buffers, because, theoretically, the latter can produce unexpected interactions with antifungal agents. One buffer that has been found to be satisfactory for antifungal testing is MOPS [3-(N-morpholino) propanesulfonic acid] (final concentration 0.165 mol/L for pH 7.0). The pH of each batch of medium is to be checked with a pH meter immediately after the medium is prepared; the pH should be between 6.9 and 7.1 at room temperature (25 °C). MIC performance characteristics of each batch of broth are evaluated using a standard set of quality control organisms."

7. PLOS authors have the option to publish the peer review history of their article (what does this mean?). If published, this will include your full peer review and any attached files.

Reviewer #1: No

Reviewer #2: No

---

## [Author Response · Author response to Decision Letter 2]

15 Sep 2021

Hedera rhombea inhibits the biofilm formation of Candida, thereby increases the susceptibility to antifungal agent, and reduces infection.

We thank the editors and the reviewers for their thoughtful and helpful comments. We have addressed, in a point-by-point manner, all the suggestions and queries from the journal and the reviewer and marked with red in manuscript. The input from the reviewers has allowed us to improve the clarity and quality of our paper. We have included below our point-by-point response to the reviewers’ comments and have included these additions and alterations to the revised manuscript.

Reviewers' comments:

Reviewer's Responses to Questions

Comments to the Author

1. If the authors have adequately addressed your comments raised in a previous round of review and you feel that this manuscript is now acceptable for publication, you may indicate that here to bypass the “Comments to the Author” section, enter your conflict-of-interest statement in the “Confidential to Editor” section, and submit your "Accept" recommendation.

Reviewer #1: (No Response)

Reviewer #2: (No Response)

2. Is the manuscript technically sound, and do the data support the conclusions?

Reviewer #1: Yes

Reviewer #2: Yes

3. Has the statistical analysis been performed appropriately and rigorously?

Reviewer #1: Yes

Reviewer #2: Yes

4. Have the authors made all data underlying the findings in their manuscript fully available?

Reviewer #1: Yes

Reviewer #2: Yes

5. Is the manuscript presented in an intelligible fashion and written in standard English?

Reviewer #1: Yes

Reviewer #2: Yes

6. Review Comments to the Author

Reviewer #1: I appreciate the changes made by the authors and I have only a handful of minor edits remaining.

Line 57 should read “The biofilm matrix acts to structure microbial communities and includes sessile cells that are frequently much more resistant to antifungal agents.”

: Thanks for your suggestion. We have modified the introduction of the problematic part.

The formation of biofilms involves multiple interconnected signaling pathways [6, 7, 8, 9, 10, 11, 12, 13, 14], and is a finely controlled process that involves attachment to surface and embedment in the exopolymer extracellular matrix [15, 16, 17, 18]. The biofilm matrix acts to structure microbial communities and includes sessile cells that are frequently much more resistant to antifungal agents. In fact, adherent C. albicans cells without specific drug resistant gene expression are up to 1,000 times more resistant to common antifungal agents than planktonic cells [19]. Therefore, the biofilm of C. albicans is a reservoir of viable fungal cells that can potentially cause systemic infections, with a mortality rate of around 40-60% [20, 21].

Line 74 I think “and reducing fungal infection” needs to be removed since virulence wasn’t assessed in the manuscript.

: Thanks for your suggestion. Based on your advice, we have changed word.

In this study, H. rhombea extract showed anti-biofilm formation activity against several fungi including C. albicans, C. tropicalis, C. glabrata, and C. parapsilosis. Interestingly, the activity of the extract also increased susceptibility to antibiotics. These results suggest that H. rhombea extract can be used as a potential anti-fungal adjuvant to control the biofilm-related infection.

Line 125 should read “nonadherent”

: Thanks for your constructive suggestions. We have made corrections in that regard based on your advice.

A human epithelial keratinocyte HaCaT cells were maintained in DMEM supplemented with 10% fetal bovine serum at 37°C in a humidified atmosphere of 5% CO2. 

HaCaT cells (0.5×106 cells per wall) were grown to confluence on 24 well plates for 24 h. DMEM were drained and then plates were cautiously washed three times with PBS to remove nonadherent cell.

Line 179 remove “was applied”

: Thank you very much. Based on your advice, we have changed word.

H. rhombea extract increased the susceptibility of antifungal agents against C. albicans.

Line 196 should just read “virulence” not “virulence-mediated”

: Thanks for your suggestion. Modified according to your advice.

The yeast-to-hyphae conversion is an important virulence property of C. albicans.

Line 290 I think this should read ”Candida albicans biofilm formation is determined by various transcription factors including BCR1, EFG1, TEC1, and NDT80 that function as components in several pathways and also influence adherence of Candida, suggesting that even though biofilm formation was initially tested, adherence or infection factors could be also regulated by treatment of H. rhombea extract.”

: Thank you very much. We have made corrections in that regard based on your advice.

C. albicans biofilm formation is determined by various transcription factors including BCR1, EFG1, TEC1, and NDT80 that function as components in several pathways and influence adherence of Candida, suggesting that even though biofilm formation was initially tested, adherence or infection factors could be also regulated by treatment of H. rhombea extract.

Figure 1 Legend has one last “fatal” instead of “fetal”

: Thank you for your valuable suggestions. We have modified

The biofilm formation of (A) C. albicans, (B) C. tropicalis var. tropicalis, (C) C. parapsilosis var. parapsilosis, (D) C. glabrata, (E) C. tropicalis, and (F) C. parapsilosis was induced in YPD media supplemented with 10% fetal bovine serum with the indicated concentrations of H. rhombea extract at 37 ℃ for 24h.

Figure 5 be sure that in the figure the label is “planktonic”.

: Thank you for your careful review. The part you mentioned has been removed.

Reviewer #2: This revised manuscript is much improved, I thank the authors for their

hard work. My main remaining concern and is with regards to the RPMI based experiments. 

The authors provided references to respond to this question. The frontiers paper that was referenced clearly describes the RPMI they used as supplemented with L-glutamine and buffered with MOPS to pH 7. (see extract 1 below..copied from the article).

The other two papers reference the CLSI standard and then they describe RPMI without sodium barcarbonate at pH7. The CLSI standard recommends the use of RPMI without sodium bicarbonate and with glutamine...but it also says to use a buffer and recommends mops (see second extract below).

If this is accurate for the experiments described here then a similar line should be added to methods. If not perhaps the authors could include a line about the pH of their extract. Or comment on the color of the media as the phenol red is in RPMI as a pH indicator.

Alternatively RPMI can use a sodium bicarbonate/Co2 buffer system....but that is usually for cell culture.

[extract 1] taken from Front. Microbiol., 15 November 2019 | https://doi.org/10.3389/fmicb.2019.02642 "A round-bottom 96-well microplate (Corning Inc., Corning, NY, United States) was filled with RPMI medium supplemented with L-glutamine (Cellgro; Corning, United States) and buffered with 165 mM morpholinepropanesulfonic acid (MOPS) (Thermo Fisher Scientific Inc., United States) at pH 7.0 "

[extract 2] taken from CLSI M27 standard for antifungal testing: "Media should be buffered to a pH of 7.0 ± 0.1 at 25 °C. A buffer should be selected that does not antagonize antifungal agents. Tris buffer is unsatisfactory, because it antagonizes the activity of flucytosine. Zwitterion buffers are preferable to buffers that readily traverse the cell membrane, such as phosphate buffers, because, theoretically, the latter can produce unexpected interactions with antifungal agents. One buffer that has been found to be satisfactory for antifungal testing is MOPS [3-(N-morpholino) propanesulfonic acid] (final concentration 0.165 mol/L for pH 7.0). The pH of each batch of medium is to be checked with a pH meter immediately after the medium is prepared; the pH should be between 6.9 and 7.1 at room temperature (25 °C). MIC performance characteristics of each batch of broth are evaluated using a standard set of quality control organisms."

: Thank you for your valuable suggestions. We have modified

C. albicans were grown overnight in YPD medium. 1 × 106 cells/mL of Candida with or without extracts were incubated in RPMI 1640 medium and YPD media supplemented with 10% fetal bovine serum medium to induce dimorphic transition at 37°C for 4 h. RPMI without sodium barcarbonate and with glutamine buffered with MOPS [3-(N-morpholino) propanesulfonic acid] to pH 7. Inhibition quantification of the yeast-to-hyphal-form transition was accomplished by counting the number of hyphae cells in the population as previously described [26, 32, 33, 34, 35, 36, 37, 38, 39, 40]. More than 1,000 cells were counted for each well in duplicate, and all assays were repeated five times. Representative results of images were obtained using a fluorescence microscope (EVOS® FL, ThermoFisher Scientific, Waltham, MA, USA). The experiments were performed in triplicate.

---

## [Editor Report · Decision Letter 3]

20 Sep 2021

Hedera rhombea inhibits the biofilm formation of Candida, thereby increases the susceptibility to antifungal agent, and reduces infection.

PONE-D-21-06872R3

Dear Dr. kim,

We’re pleased to inform you that your manuscript has been judged scientifically suitable for publication and will be formally accepted for publication once it meets all outstanding technical requirements.

Kind regards,

Roy Aziz Khalaf

Academic Editor

PLOS ONE

Additional Editor Comments (optional):

All requested revisions were performed
---

## [Editor Report · Acceptance letter]

27 Sep 2021

PONE-D-21-06872R3 

*Hedera rhombea* inhibits the biofilm formation of *Candida*, thereby increases the susceptibility to antifungal agent, and reduces infection. 

Dear Dr. Kim:

I'm pleased to inform you that your manuscript has been deemed suitable for publication in PLOS ONE. Congratulations! Your manuscript is now with our production department. 

Kind regards, 

on behalf of

Dr. Roy Aziz Khalaf 

Academic Editor

PLOS ONE